# PRE-TRAINING BY COMPLETING POINT CLOUDS

## ABSTRACT

There has recently been a flurry of exciting advances in deep learning models on point clouds. However, these advances have been hampered by the difficulty of creating labelled point cloud datasets: sparse point clouds often have unclear label identities for certain points, while dense point clouds are time-consuming to annotate. Inspired by mask-based pre-training in the natural language processing community, we propose a pre-training mechanism based point clouds completion. It works by masking occluded points that result from observations at different camera views. It then optimizes a completion model that learns how to reconstruct the occluded points, given the partial point cloud. In this way, our method learns a pre-trained representation that can identify the visual constraints inherently embedded in real-world point clouds. We call our method *Occlusion Completion* (OcCo). We demonstrate that OcCo learns representations that improve the semantic understandings as well as generalization on downstream tasks over prior methods, transfer to different datasets, reduce training time and improve label efficiency.

## 1 INTRODUCTION

Point clouds are a natural representation of 3D objects. Recently, there has been a flurry of exciting new point cloud models in areas such as segmentation (Landrieu & Simonovsky, 2018; Yang et al., 2019a; Hu et al., 2020a) and object detection (Zhou & Tuzel, 2018; Lang et al., 2019; Wang et al., 2020b). Current 3D sensing modalities (*i.e.*, 3D scanners, stereo cameras, lidars) have enabled the creation of large repositories of point cloud data (Rusu & Cousins, 2011; Hackel et al., 2017). However, annotating point clouds is challenging as: (1) Point cloud data can be sparse and at low resolutions, making the identity of points ambiguous; (2) Datasets that are not sparse can easily reach hundreds of millions of points (e.g., small dense point clouds for object classification (Zhou & Neumann, 2013) and large vast point clouds for 3D reconstruction (Zolanvari et al., 2019)); (3) Labelling individual points or drawing 3D bounding boxes are both more complex and time-consuming compared with annotating 2D images (Wang et al., 2019a). Since most methods require dense supervision, the lack of annotated point cloud data impedes the development of novel models.

On the other hand, because of the rapid development of 3D sensors, unlabelled point cloud datasets are abundant. Recent work has developed unsupervised pre-training methods to learn initialization for point cloud models. These are based on designing novel generative adversarial networks (GANs) (Wu et al., 2016; Han et al., 2019; Achlioptas et al., 2018) and autoencoders (Hassani & Haley, 2019; Li et al., 2018a; Yang et al., 2018). However, completely unsupervised pre-training methods have been recently outperformed by the self-supervised pre-training techniques of (Sauder & Sievers, 2019) and (Alliegro et al., 2020). Both methods work by first voxelizing point clouds, then splitting each axis into $k$ parts, yielding $k^3$ voxels. Then, voxels are randomly permuted, and a model is trained to rearrange the permuted voxels back to their original positions. The intuition is that such a model learns the spatial configuration of objects and scenes. However, such random permutation destroys all spatial information that the model could have used to predict the final object point cloud.

Our insight is that partial point-cloud masking is a good candidate for pre-training in point-clouds because of two reasons: (1) The pre-trained model requires spatial and semantic understanding of the input point clouds to be able to reconstruct masked shapes. (2) Mask-based completion tasks have become the *de facto* standard for learning pre-trained representations in natural language processing (NLP) (Mikolov et al., 2013; Devlin et al., 2018; Peters et al., 2018). Different from random permutations, masking respects the spatial constraints that are naturally encoded in point clouds of real-world objects and scenes. Given this insight, we propose *Occlusion Completion* (OcCo)

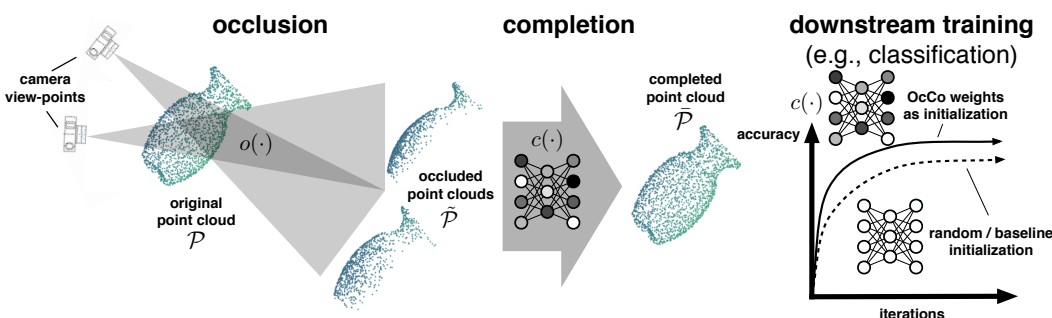

Figure 1: OcCo consists of two steps: (a) occlusion $o(\cdot)$ of a point cloud $\mathcal{P}$ based on a random camera view-point into a partial point cloud $\tilde{\mathcal{P}}$, and (b) a model $c(\cdot)$ that completes the occluded point cloud $\bar{\mathcal{P}}$ so that $\bar{\mathcal{P}} \approx \mathcal{P}$. We demonstrate that the completion model $c(\cdot)$ can be used as initialization for downstream tasks, leading to faster training and better generalization over existing methods.

a self-supervised pre-training method that consists of (a) a mechanism to generate occluded point clouds, and (b) a completion task to reconstruct the occluded point cloud.

Specifically, in (a) point clouds are generated by determining what part of objects would be occluded if the underlying object was observed from a particular view-point. In fact, many point clouds generated from a fixed 3D sensor will have occlusions exactly like this. Given an occluded point cloud, the goal of the completion task (b) is to learn a model that accurately reconstructs the missing parts of the point cloud. For a model to perform this task well, it needs to learn to encode localized structural information, based on the context and geometry of partial objects. This is something that is useful for any point cloud model to know, even if used only for classification or segmentation.

We demonstrate that the weights learned by our pre-training method *on a single unsupervised dataset* can be used as initialization for models in downstream tasks (e.g., object classification, part and semantic segmentation) to improve them, *even on completely different datasets*. Specifically our pre-training technique: (i) leads to improved generalization over prior baselines on the downstream tasks of object classification, object part and scene semantic segmentation; (ii) speeds up model convergence, in some cases, by up to $5\times$; (iii) maintains improvements as the size of the labelled downstream dataset decreases; (iv) can be used for a variety of state-of-the-art point cloud models.

## 2    OCCLUSION COMPLETION

We now introduce Occlusion Completion (OcCo). Our approach is shown in Figure 1. Our main insight is that by continually occluding point clouds and learning a model $c(\cdot)$ to complete them, the weights of the completion model can be used as initialization for downstream tasks (e.g., classification, segmentation) , speeding up training and improving generalization over other initialization techniques.

Throughout we assume point clouds $\mathcal{P}$ are sets of points in 3D Euclidean space, $\mathcal{P} = \{p_1, p_2, ..., p_n\}$, where each point $p_i$ is a vector of coordinates $(x_i, y_i, z_i)$ and features (e.g. color and normal). We begin by describing the components that make up our occlusion mapping $o(\cdot)$. Then we detail how to learn a completion model $c(\cdot)$, giving pseudocode and the architectural details in appendix. Finally we discuss the criteria on validating the effectiveness of a pre-training model for 3D point clouds.

### 2.1    GENERATING OCCLUSIONS

We first describe a randomized occlusion mapping $o : \mathbb{P} \to \mathbb{P}$ (where $\mathbb{P}$ is the space of all point clouds) from a full point cloud $\mathcal{P}$ to an occluded point cloud $\tilde{\mathcal{P}}$. We will do so by determining which points are occluded when the point cloud is viewed from a particular camera position. This requires three steps: (1) A projection of the point cloud (in a world reference frame) into the coordinates of a camera reference frame; (2) Determining which points are occluded based on the camera view-points; (3) Mapping the points back from the camera reference frame to the world reference frame.

**Viewing the point cloud from a camera.**    A camera defines a projection from a 3D world reference frame into a distinctive 3D camera reference frame. It does so by specifying a camera model and a camera view-point from which the projection occurs. The simplest camera model is the pinhole

camera, and view-point projection for it is given by a simple linear equation:

$$
\begin{bmatrix} x_{\text{cam}} \\ y_{\text{cam}} \\ z_{\text{cam}} \end{bmatrix} = \underbrace{\begin{bmatrix} f & \gamma & w/2 \\ 0 & f & h/2 \\ 0 & 0 & 1 \end{bmatrix}}_{\substack{\text{intrinsic} \\ [\ \mathbf{K}\ ]}} \underbrace{\left[\begin{array}{ccc|c} r_1 & r_2 & r_3 & t_1 \\ r_4 & r_5 & r_6 & t_2 \\ r_7 & r_8 & r_9 & t_3 \end{array}\right]}_{\substack{\text{rotation}\ |\ \text{translation} \\ [\ \mathbf{R}\ |\ \mathbf{t}\ ]}} \begin{bmatrix} x \\ y \\ z \\ 1 \end{bmatrix}.
\tag{1}
$$

In the above, $(x, y, z)$ are the original point cloud coordinates, the matrix including $r$ and $t$ entries is the concatenation of a 3D rotation matrix with a 3D translation vector, and the final matrix to the left is the camera intrinsic matrix ($f$ specifies the camera focal length, $\gamma$ is the skewness between the $x$ and $y$ axes in the camera, and $w, h$ are the width and height of the camera image). Given these, the final coordinates $(x_{\text{cam}}, y_{\text{cam}}, z_{\text{cam}})$ are the positions of the point in the camera reference frame. We will refer to the intrinsic matrix as $\mathbf{K}$ and the rotation/translation matrix as $[\mathbf{R}|\mathbf{t}]$.

**Determining occluded points.** We can think of the point $(x_{\text{cam}}, y_{\text{cam}}, z_{\text{cam}})$ in multiple ways: (a) as a 3D point in the camera reference frame, (b) as a 2D pixel with coordinates $(f x_{\text{cam}}/z_{\text{cam}}, f y_{\text{cam}}/z_{\text{cam}})$ with a depth of $z_{\text{cam}}$. In this way, some 2D points resulting from the projection may be occluded by others if they have the same pixel coordinates, but appear at a larger depth. To determine which points are occluded, we first use *Delaunay triangulation* to reconstruct a polygon mesh from the points, and remove the points which belong to the hidden surfaces that are determined via *z-buffering*.

**Mapping back from camera frame to world frame.** Once occluded points are removed, we re-project the point cloud to the original world reference frame, via the following linear transformation:

$$
\begin{bmatrix} x' \\ y' \\ z' \\ 1 \end{bmatrix} = \underbrace{\left[\begin{array}{ccc|c} r_1 & r_2 & r_3 & t_1 \\ r_4 & r_5 & r_6 & t_2 \\ r_7 & r_8 & r_9 & t_3 \\ 0 & 0 & 0 & 1 \end{array}\right]^{\top}}_{\left[\begin{smallmatrix} \mathbf{R}\ |\ \mathbf{t} \\ \mathbf{0}\ |\ 1 \end{smallmatrix}\right]^{\top}} \underbrace{\begin{bmatrix} 1/f & -\gamma/f^2 & (\gamma h - fw)/(2f^2) & 0 \\ 0 & 1/f & -h/(2f) & 0 \\ 0 & 0 & 1 & 0 \\ 0 & 0 & 0 & 1 \end{bmatrix}}_{\left[\begin{smallmatrix} \mathbf{K}^{-1} & \mathbf{0} \\ \mathbf{0} & 1 \end{smallmatrix}\right]} \begin{bmatrix} x_{\text{cam}} \\ y_{\text{cam}} \\ z_{\text{cam}} \\ 1 \end{bmatrix}.
\tag{2}
$$

Our randomized occlusion mapping $o(\cdot)$ is constructed as follows. Fix an initial point cloud $\mathcal{P}$. Given a camera intrinsics matrix $\mathbf{K}$, sample rotation/translation matrices $[[\mathbf{R}_1|\mathbf{t}_1], \ldots, [\mathbf{R}_V|\mathbf{t}_V]]$, where $V$ is the number of views. For each view $v \in [V]$, project $\mathcal{P}$ into the camera frame of that view-point using eq. (1), find occluded points and remove them, then map the rest back to the world reference using eq. (2). This yields the final occluded world frame point cloud for view-point $v$: $\tilde{\mathcal{P}}_v$.

## 2.2 The Completion Task

Given an occluded point cloud $\tilde{\mathcal{P}}$ produced by $o(\cdot)$, the goal of the completion task is to learn a completion mapping $c : \mathbb{P} \to \mathbb{P}$ from $\tilde{\mathcal{P}}$ to a completed point cloud $\overline{\mathcal{P}}$. We say that a completion mapping is accurate w.r.t. loss $\ell(\cdot, \cdot)$ if $\mathbb{E}_{\tilde{\mathcal{P}} \sim o(\mathcal{P})} \ell(c(\tilde{\mathcal{P}}), \mathcal{P}) \to 0$. The structure of the completion model $c(\cdot)$ is an "encoder-decoder" network (Dai et al., 2017b; Yuan et al., 2018; Tchapmi et al., 2019; Wang et al., 2020a). The encoder maps an occluded point cloud to a vector, and the decoder reconstructs the full shape. After pre-training, the encoder weights can be used as initialization for downstream tasks. In appendix we gives pseudocode for OcCo and describes the architectures.

## 3 Experiments

### 3.1 Pre-Training and Downstream Training Details

We evaluate how OcCo improves the learning and generalization of a number of classification and segmentation tasks. Here we describe the details of training in each setting.

**OcCo pre-training.** For all experiments, we will use a single pre-training dataset based on ModelNet40 (Wu et al., 2015). It includes 12,311 synthesized objects from 40 object categories, divided into 9,843 training objects and 2,468 testing objects. To construct the pre-training dataset,

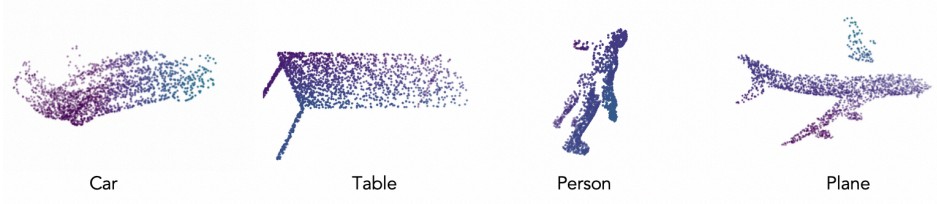

Figure 2: Examples of self-occluded objects generated by our method.

we generate occluded point clouds based on the training objects with a fixed camera intrinsics $\{f = 1000, \gamma = 0, \omega = 1600, h = 1200\}$, 10 random selected viewpoints and zero translation. Figure 2 shows examples of the resulting occluded point clouds. Given these, we train an "encoder-decoder" style completion model $c(\cdot)$. For encoders, similar to prior completion models (Tchapmi et al., 2019; Wang et al., 2020a; Wen et al., 2020a), we consider PointNet (Qi et al., 2017a), PCN (Yuan et al., 2018) and DGCNN (Wang et al., 2019b). These networks encode an occluded point cloud into a 1024-dimensional vector. We adapted the folding-based decoder from (Yuan et al., 2018) to complete the point clouds in a two-stage procedure. We use the Chamfer Distance (CD) as our loss function $\ell(\cdot, \cdot)$. We use Adam (Kingma & Ba, 2015) with an initial learning rate of 1e-4, decayed by 0.7 every 10 epochs to a minimum value of 1e-6, for a total of 50 epochs. We use a batch size of 32 and set the momentum in the batch normalisation to be 0.9.

**Few-shot learning.** We use ModelNet40 and Syndey10 (De Deuge et al., 2013) for "$K$-way $N$-shot" learning. During training, $K$ classes are randomly selected and for each class we sample $N$ random samples, then the model is tested on the same $K$ classes . As in Sharma & Kaul (2020), we represent each object with 100 points. We use the same training settings as used in the next paragraph.

**Object classification.** We use three 3D object recognition benchmarks: ModelNet40, Scan-Net10 (Qin et al., 2019) and ScanObjectNN (Uy et al., 2019); we describe them in the appendix. All objects are represented with 1024 points. We use the same training settings as the original works. Concretely, for PCN and PointNet, we use the Adam optimizer with an initial learning rate 1e-3, decayed by 0.7 every 20 epochs to a minimum value of 1e-5. For DGCNN, we use the SGD optimizer with a momentum of 0.9 and a weight decay of 1e-4. The learning rate starts from 0.1 and then reduces using cosine annealing Loshchilov & Hutter (2017) with a minimum value of 1e-3. We use dropout Srivastava et al. (2014) in the fully connected layers before the softmax output layer. The dropout rate of PointNet and PCN is set to 0.7, and is 0.5 for DGCNN. For all three models, we train them for 200 epochs with a batch size of 32. We report the results based on three runs.

**Part segmentation.** We use the ShapeNetPart (Armeni et al., 2016) benchmark for object part segmentation. This dataset contains 16,881 objects from 16 categories, and has 50 parts in total. Each object is represented with 2048 points, and we use the same training settings as the original work.

**Semantic segmentation.** We use the S3DIS benchmark (Armeni et al., 2016) for semantic indoor scene segmentation. It contains 3D scans collected via Matterport scanners in 6 different places, encompassing 271 rooms. Each point, described by a 9-dimensional vector (including coordinates, RGB values and normalised location), is labeled as one of 13 semantic categories (e.g. chair, table and floor). We use the same preprocessing procedures and training settings as the original work.

### 3.2 What is Learned from OcCo Pre-Training

Alongside OcCo's ability to improve learning tasks we analyze the properties of the pre-trained representation itself. Here we describe the approaches we use for such analyses.

**Visualisation of learned features.** *Feature visualisation* (Olah et al., 2017) is widely used to qualitatively understand the role of a convolutional neural network unit. It links highly activated parts of a CNN channel with human concepts which have semantic meaning. Ideally the pre-training process learns disentangled features that are useful to distinguish different parts of an object or a scene. These learned features will be beneficial to not only few-shot learning, but also object recognition and part and scene segmentation.

**Detection of semantic concepts.**   To quantitatively analyse the learned features of pre-training, we adapt *network dissection* (Bau et al., 2017; 2020) to determine the number of concept detectors in a pre-trained point cloud feature encoder. Specifically, for the $k$-th channel, we first create a binary activation mask $M_k$ based on highly activated point subsets. Since point cloud encoders usually learn each point feature either independently or via neighborhood aggregation, the feature maps usually do not change in the vertical direction (see Figure 3). Therefore we can skip the *retrieval* step and directly quantify the alignment between an activation mask $M_k$ and the $n$-th concept mask $C_n$ (i.e., object parts) via mean intersection of union (mIoU) over a collection of point clouds $\mathcal{D}_\mathcal{P}$:

$$\text{mIoU}_{(k,n)} = \mathbb{E}_{\mathcal{P} \sim \mathcal{D}_\mathcal{P}} \left[ \frac{|M_k(\mathcal{P}) \cap C_n(\mathcal{P})|}{|M_k(\mathcal{P}) \cup C_n(\mathcal{P})|} \right] \tag{3}$$

where $|\cdot|$ is the set cardinality. $\text{mIoU}_{(k,n)}$ can be interpreted as how well unit $k$ detects concept $c$.

**Structural invariance/equivariance under SO(3) transformation.**   Pre-training should learn a representation that is robust under under rigid SO(3) transformations (i.e., rotation, translation, permutation). Although a single representation might vary after transformation, the cluster structures should be preserved. We use adjusted mutual information (AMI) (Nguyen et al., 2009) based on the clustering $\Omega$ and the ground truth label $\mathcal{C}$, which prevents the score from monotonically increasing when the number of clusters increases,

$$\text{AMI}(\Omega, \mathcal{C}) = \mathbb{E}_{\mathcal{P} \sim \mathcal{D}_\mathcal{P}} \left[ \frac{\text{I}(\Omega; \mathcal{C}) - \mathbb{E}[\text{I}(\Omega; \mathcal{C})]}{(\text{H}(\Omega) + \text{H}(\mathcal{C}))/2 - \mathbb{E}[\text{I}(\Omega; \mathcal{C})]} \right] \tag{4}$$

where $\Omega$ is the clustering determined by the learned embeddings $\text{Enc}(\cdot)$ and unsupervised clustering methods such as K-means. $\text{I}(\Omega; C) = \sum_k \sum_j P(w_k \cap c_j) \log \frac{P(w_k \cap c_j)}{P(w_k)P(c_j)}$ denotes the mutual information, $\text{H}(\cdot)$ is the entropy. AMI has a maximal of 1 when two partitions are identical, and reaches a minimal of 0 if two clusters are total uncorrelated. It is calculated as:

$$\mathcal{L} = \mathbb{E}_{\mathcal{P} \sim \mathcal{D}_\mathcal{P}, \mathcal{S} \sim \text{SO}(3)} [\text{AMI}(\Omega, \text{Enc}(\mathcal{S}(\mathcal{P})))] \tag{5}$$

Once we finish the pre-training on ModelNet40, we first analyse the learned features and embeddings of the OcCo PointNet via the tests or *probes* described above. Specifically, we examine the learned concepts of the pre-trained encoders on ShapeNetPart. We assign activation mask $M_k$ with points that have top 20% highest values in the $k$-th unit of the feature, and the $n$-th concept mask $C_n$ is derived from the ground-truth annotations of the $n$-th object parts. We ignore the object parts which have less than 100 points. We call $k$-th channel a detector of concept $n$ when $\text{mIoU}_{(k,n)} > 0.5$.

We analyze the learned embeddings of the pre-trained encoders on ShapeNet10 and ScanObjectNN. We cluster the learned embeddings $\text{Enc}(\mathcal{P})$ into $\Omega$ with K-means and calculate the AMI w.r.t labels. Since the encoders are permutation invariant, here we consider rotation, translation and jittering.

### 3.3   Completion Results and Probe Tests

**Visualisation of learned features.**   In Figure 3, we first visualize the features learned by OcCo PointNet on the objects from test split of ModelNet40. We visualize each learned feature by

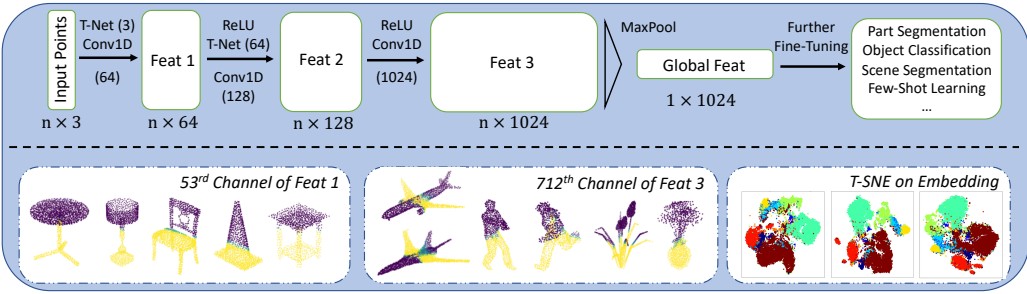

Figure 3: Visualisation on the learned features and embeddings of OcCo-initialised encoders. Above half illustrates the location of learned features in the architecture of PointNet (Qi et al., 2017a).

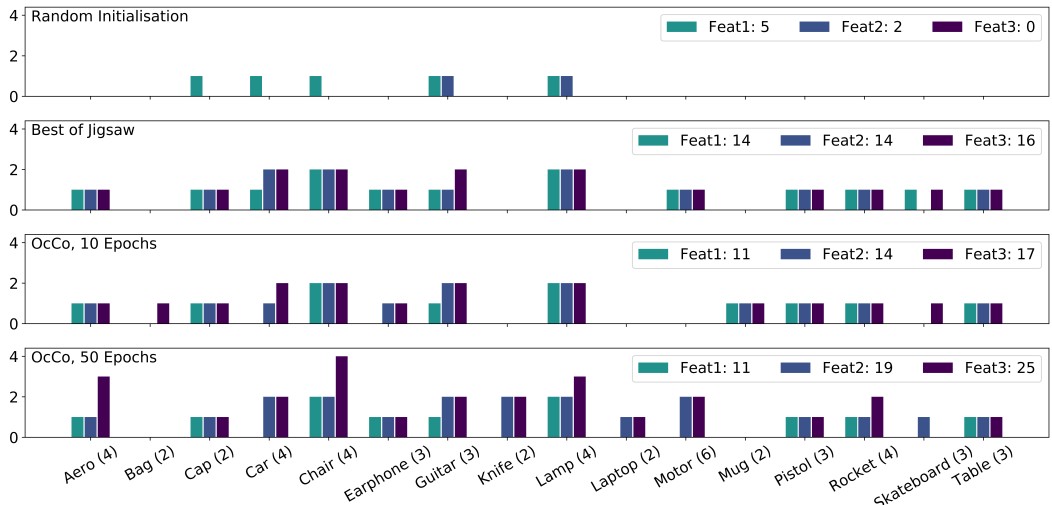

Figure 4: Number of (unique) detected object parts in the feature maps of random, Jigsaw and OcCo-initialised PointNet. Digit in the bracket is the number of parts in that object category.

coloring the points according to their channel values. We find that, in early stage the encoder is able to learn low-level geometric primitives, i.e., planes, cylinders and cones, while later the network recognises more complex shapes like wings, leafs and upper bodies. We use t-SNE on the embeddings of OcCo encoders based on ShapeNet10, distinguishable clusters are formed for different categories.

**Number of concept detectors.** In Figure 4, we sketch the number of detected parts based on random, Jigsaw and OcCo (trained for 10 epochs and 50 epochs)-initialised PointNet. We find that, while keeping the previously learned concepts, OcCo helps the encoder progressively detect more object parts as the training proceeds. We show that OcCo have outperformed prior methods in terms of total detected parts (numbers in legends). We provide visualisations in the appendix.

Table 1: Adjusted mutual information (AMI) under transformations. We reported the mean and std over 10 runs. 'J', 'T', 'R' stand for jittering, translation and rotation respectively.

| Transformation | | | ShapeNet10 | | | | ScanObjectNN | | | |
|---|---|---|---|---|---|---|---|---|---|---|
| J | T | R | VFH | M2DP | Jigsaw | OcCo | VFH | M2DP | Jigsaw | OcCo |
| | | | 0.12±0.01 | 0.22±0.03 | 0.33±0.04 | **0.51±0.03** | 0.05±0.02 | 0.18±0.02 | 0.29±0.02 | **0.44±0.03** |
| ✓ | | | 0.12±0.02 | 0.19±0.02 | 0.32±0.02 | **0.45±0.02** | 0.06±0.02 | 0.17±0.02 | 0.27±0.02 | **0.42±0.04** |
| ✓ | ✓ | | 0.13±0.03 | 0.21±0.02 | 0.29±0.07 | **0.38±0.04** | 0.04±0.02 | 0.18±0.03 | 0.24±0.04 | **0.39±0.06** |
| ✓ | ✓ | ✓ | 0.07±0.03 | 0.20±0.04 | 0.28±0.03 | **0.35±0.05** | 0.04±0.01 | 0.16±0.03 | 0.18±0.09 | **0.34±0.06** |

**Invariance/Equivariance under SO(3) transformation.** We compare Jigsaw and OcCo-initialised PointNet encoder with two hand-crafted point cloud global descriptors: viewpoint feature histogram (VFH) (Rusu et al., 2010) and M2DP (He et al., 2016) in Table 1. Each point cloud is represented as a vector, and we use K-means for clustering, where K is set as the number of categories. We show that OcCo pre-training helps the networks to learn better embeddings of point cloud objects, especially when they are occluded and with outlier points (ScanObjectNN).

## 3.4 FEW-SHOT LEARNING

We use the same setting and train/test split as (Sharma & Kaul, 2020) (cTree), and report the mean and standard deviation across on 10 runs. The top half of the table reports results for eight randomly initialized point cloud models, while the bottom-half reports results on two models across three pre-training methods. We **bold** the best results (and those whose standard deviation overlaps the mean of the best result). It is worth mentioning (Sharma & Kaul, 2020) pre-trained the encoders on both datasets before fine tuning, while we only pre-trained once on ModelNet40. The results show that models pre-trained with OcCo either outperform or have standard deviations that overlap with the best method in 7 out of 8 settings.

Table 2: Few-shot classification accuracy

| Baseline | ModelNet40 | | | | Sydney10 | | | |
|---|---|---|---|---|---|---|---|---|
| | 5-way | | 10-way | | 5-way | | 10-way | |
| | 10-shot | 20-shot | 10-shot | 20-shot | 10-shot | 20-shot | 10-shot | 20-shot |
| 3D-GAN | 55.8±10.7 | 65.8±9.9 | 40.3±6.5 | 48.4±5.6 | 54.2±4.6 | 58.8±5.8 | 36.0±6.2 | 45.3±7.9 |
| FoldingNet | 33.4 ±13.1 | 35.8±18.2 | 18.6±6.5 | 15.4±6.8 | 58.9±5.6 | 71.2±6.0 | 42.6±3.4 | 63.5±3.9 |
| Latent-GAN | 41.6±16.9 | 46.2±19.7 | 32.9±9.2 | 25.5±9.9 | 64.5±6.6 | 79.8±3.4 | 50.5±3.0 | 62.5±5.1 |
| PointCapsNet | 42.3±17.4 | 53.0±18.7 | 38.0±14.3 | 27.2±14.9 | 59.4±6.3 | 70.5±4.8 | 44.1±2.0 | 60.3±4.9 |
| PointNet++ | 38.5±16.0 | 42.4±14.2 | 23.1±7.0 | 18.8±5.4 | **79.9±6.8** | 85.0±5.3 | 55.4±2.2 | 63.4±2.8 |
| PointCNN | 65.4±8.9 | 68.6±7.0 | 46.6±4.8 | 50.0±7.2 | 75.8±7.7 | 83.4±4.4 | 56.3±2.4 | 73.1±4.1 |
| PointNet, Rand | 52.0±12.2 | 57.8±15.5 | 46.6±13.5 | 35.2±15.3 | 74.2±7.3 | 82.2±5.1 | 51.4±1.3 | 58.3±2.6 |
| PointNet, cTree | 63.2±10.7 | 68.9±9.4 | 49.2±6.1 | 50.1±5.0 | 76.5±6.3 | 83.7±4.0 | 55.5±2.3 | 64.0±2.4 |
| PointNet, OcCo | **89.7±6.1** | **92.4±4.9** | **83.9±5.6** | **89.7±4.6** | 77.7±8.0 | 84.9±4.9 | 60.9±3.7 | 65.5±5.5 |
| DGCNN, Rand | 31.6 ±9.0 | 40.8±14.6 | 19.9±6.5 | 16.9±4.8 | 58.3±6.6 | 76.7±7.5 | 48.1±8.2 | 76.1±3.6 |
| DGCNN, Jigsaw | 34.3±4.1 | 42.2±11.0 | 26.0±7.5 | 29.9±8.2 | 52.5±6.6 | 79.6±6.0 | 52.7±3.3 | 69.1±2.6 |
| DGCNN, cTree | 60.0±8.9 | 65.7±8.4 | 48.5±5.6 | 53.0±4.1 | **86.2±4.4** | **90.9±2.5** | **66.2±2.8** | **81.5±2.3** |
| DGCNN, OcCo | **90.6±2.8** | **92.5±6.0** | **82.9±4.1** | **86.5±7.1** | 79.9±6.7 | 86.4±4.7 | 63.3±2.7 | 77.6±3.9 |

## 3.5 OBJECT CLASSIFICATION RESULTS

We now compare OcCo against prior initialization approaches on object classification tasks. Table 3 compares OcCo-initialization to random (Rand) and (Sauder & Sievers, 2019)'s (Jigsaw) initialization on various object classification datasets among different encoders. "MN40", "ScN10" and "SO15" stand for ModelNet40, ScanNet10 and ScanObjectNN respectively. Recall that OcCo-initialization is pre-trained only on occlusions generated from the train split of ModelNet40. We color blue the best results for each encoder and **bold** in black the overall best result (and those whose standard deviation overlaps the mean of the best result) for each dataset. We show that OcCo-initialized models outperform all baselines. These results demonstrate that the OcCo-initialized models have strong transfer capabilities on out-of-domain datasets. We make more comparisons in the appendix.

Table 3: Comparison between OcCo, Jigsaw and Rand initialization on 3D object recognition benchmarks. After confirming the scores from (Qi et al., 2017a; Wang et al., 2019b; Uy et al., 2019; Sauder & Sievers, 2019) are reproducible, we reported the mean and standard error over three runs.

| Dataset | PointNet | | | PCN | | | DGCNN | | |
|---|---|---|---|---|---|---|---|---|---|
| | Rand | Jigsaw | OcCo | Rand | Jigsaw | OcCo | Rand | Jigsaw | OcCo |
| MN40 | 89.2±0.1 | 89.6±0.1 | 90.1±0.1 | 89.3±0.1 | 89.6±0.2 | 90.3±0.2 | 92.5±0.4 | 92.3±0.3 | **93.0±0.2** |
| ScN10 | 76.9±0.2 | 77.2±0.2 | 78.0±0.2 | 77.0±0.3 | 77.9±0.3 | 78.2±0.3 | 76.1±0.7 | 77.8±0.5 | **78.5±0.3** |
| SO15 | 73.5±0.5 | 76.5±0.4 | 80.0±0.2 | 78.3±0.3 | 78.2±0.1 | 80.4±0.2 | 82.4±0.4 | 82.7±0.8 | **83.9±0.4** |

## 3.6 OBJECT PART SEGMENTATION RESULTS

Table 4 compares OcCo-initialization to random and (Sauder & Sievers, 2019)'s (Jigsaw) initialization on object part segmentation task. We show that OcCo-initialized models outperform or match others in terms of accuracy and IoU in all three encoders, demonstrating representations derived from completing occluded ModelNet40 improves the performance of part segmentation.

Table 4: Overall point prediction accuracy (mAcc) and mean intersection of union (mIoU) on ShapeNetPart. We reported the mean and standard error based on three runs.

| | PointNet | | | PCN | | | DGCNN | | |
|---|---|---|---|---|---|---|---|---|---|
| | Rand | Jigsaw | OcCo | Rand | Jigsaw | OcCo | Rand | Jigsaw | OcCo |
| mAcc | 92.8±0.5 | 93.1±0.3 | 93.4±0.4 | 92.3±0.6 | 92.6±0.5 | 93.0±0.5 | 92.2±0.5 | 92.7±0.5 | **94.4±0.4** |
| mIoU | 82.2±1.4 | 82.2±1.6 | 83.4±1.1 | 81.3±1.5 | 81.2±1.7 | 82.3±1.4 | 84.4±0.7 | **84.3±0.7** | **85.0±0.6** |

## 3.7 SEMANTIC SEGMENTATION

Here we compare random, Jigsaw and OcCo initialization on semantic segmentation task. We follow the same design of PointNet and DGCNN, use a $k$-fold train-test procedure as in (Armeni et al., 2016). The results are reported in Table 5. OcCo-initialized models outperform random and jigsaw-initialized ones, demonstrating that the pre-trained representations derived from completing occluded ModelNet40 brings improvements on segmenting indoor scenes which consist of occluded objects.[1]

---

[1] We noticed that the random initialised/pre-trained model in (Sauder & Sievers, 2019) (mIoU=40.3/41.2) did not achieve the similar results as the original DGCNN (mIoU=56.1). They consider a transductive setting which is not directly comparable to ours, so here we stick to the supervised setting and report our reproduced scores.

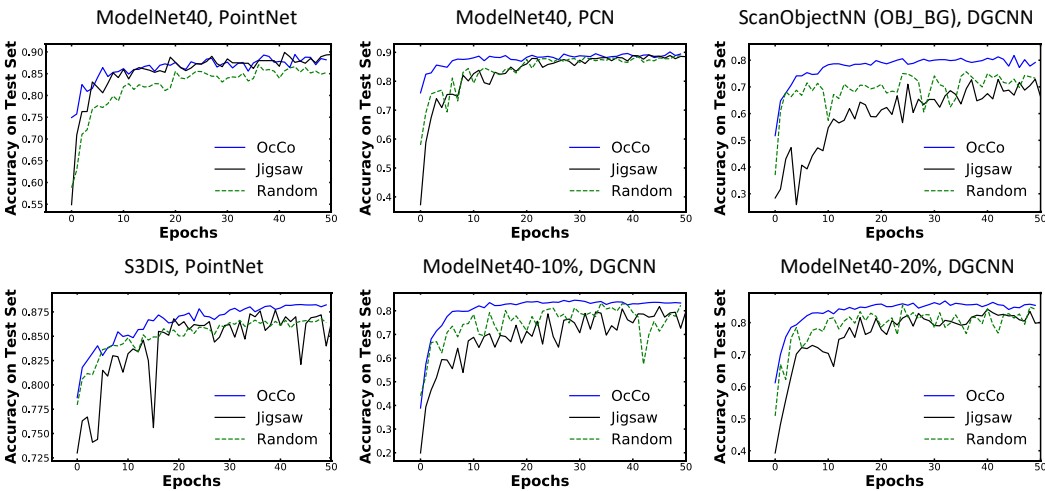

Figure 5: Learning curves of random, Jigsaw and OcCo , '10%' is the portion of used training data

Table 5: Overall point prediction accuracy (mAcc) and mean class intersection of union (mIoU) on the S3DIS averaged across 6-cv-fold over three runs. OcCo encoders are pre-trained on ModelNet40.

| | PointNet | | | PCN | | | DGCNN | | |
|---|---|---|---|---|---|---|---|---|---|
| | Rand | Jigsaw | OcCo | Rand | Jigsaw | OcCo | Rand | Jigsaw | OcCo |
| mAcc | 78.2±0.4 | 80.1±0.7 | 82.0±0.6 | 82.9±0.5 | 83.7±0.4 | **85.1±0.3** | 83.7±0.4 | 84.1±0.4 | 84.6±0.3 |
| mIoU | 47.0±0.8 | 52.6±1.1 | 54.9±0.6 | 51.1±1.4 | 52.2±1.1 | 53.4±1.2 | 54.9±1.2 | 55.6±0.8 | **58.0±1.0** |

## 3.8 LEARNING CURVES

We plot the learning curves for classification and segmentation tasks in Figure 5. We observe that the models with OcCo initialization converge faster to better test accuracy than the random and sometimes Jigsaw-initialized models. For example, on ModelNet40 with a PCN encoder, the OcCo-initialized model takes around 10 epochs to converge, while the randomly initialized model takes around 50 epochs. Similarly, for ScanObjectNN with DGCNN encoder, the OcCo-initialized model converges around 20 epochs and to a better test accuracy than the random and Jigsaw-initialized model.

## 4 RELATED WORK

### 4.1 DEEP MODELS FOR POINT CLOUDS

Work on deep models for point clouds can largely be divided into three different structural approaches: (a) *pointwise-based networks*, (b) *convolution-based networks*, and (c) *graph-based networks*. We call the networks that independently process each point, before aggregating these point representations: *pointwise-based networks* (Qi et al., 2017a;b; Joseph-Rivlin et al., 2019; Duan et al., 2019; Zhao et al., 2019; Yang et al., 2019c; Lin et al., 2019). One well-known method, PointNet, devises a novel neural network that is designed to respect the permutation invariance of point clouds. Each point is independently fed into a multi-layer perceptron, then outputs are aggregated using a permutation-invariant function (e.g., max-pooling) to obtain a global point cloud representation. Another class of methods are *convolution-based networks* (Hua et al., 2018; Su et al., 2018; Li et al., 2018b; Atzmon et al., 2018; Landrieu & Simonovsky, 2018; Hermosilla et al., 2018; Groh et al., 2018; Rao et al., 2019). These works map point clouds to regular grid structures and extend the classic convolution operator to handle these grid structures. A representative model, PCNN (Atzmon et al., 2018), defines two operators, extension and restriction, for mapping point cloud functions to volumetric functions and vise versa. The third class of models is *graph-based networks* (Simonovsky & Komodakis, 2017; Wang et al., 2019b; Shen et al., 2018; Wang et al., 2018; Zhang & Rabbat, 2018; Chen et al., 2019). These networks regard each point as a vertex of a graph and generate edges based on spatial information and node similarities. A popular method is DGCNN (Wang et al., 2019b), which introduces a new operation, EdgeConv, to aggregate local features and a graph update module to learn dynamic graph relations from layer to layer. NRS (Cao et al., 2020) uses a neural random subspace method based on the encoded embeddings to further improve the model performance.

## 4.2 Pre-Training for Point Clouds

Pre-training models on unlabelled data are gaining popularity recently due to its success on a wide range of tasks, such as natural language understanding (Mikolov et al., 2013; Devlin et al., 2018), object detection (He et al., 2020; Chen et al., 2020) and graph representations (Hu et al., 2020c;d). The representations learned from these pre-trained models can be used as a good initializer in downstream tasks, where task-specific annotated samples are scarce. The three most common pre-training objectives for point clouds are based on: (i) generative adversarial networks (GAN), (ii) autoencoders, and (iii) spatial relation (Sauder & Sievers, 2019; Sharma & Kaul, 2020). However, GANs for point clouds are limited to non-point-set inputs, i.e., voxelized representations (Wu et al., 2016), 2D depth images of point clouds (Han et al., 2019), and latent representations from autoencoders (Achlioptas et al., 2018), as sampling point sets from a neural network is non-trivial. Thus these GAN approaches cannot leverage the natural order-invariance of point-sets. Autoencoders (Yang et al., 2018; Li et al., 2018a; Hassani & Haley, 2019; Shi et al., 2020) learn to encode point clouds into a latent space before reconstructing these point clouds from their latent representation. Similar to these methods, generative models based on normalizing flow (Yang et al., 2019b) and approximate convex decomposition (Gadelha et al., 2020) have been shown effective for the unsupervised learning on point clouds. However, both GAN and autoencoder-based pre-training methods have been recently outperformed on downstream tasks by the pre-training technique of Sauder & Sievers (2019) or Sharma & Kaul (2020) in few-shot setting.

These methods are based on spatial relation reconstruction, which aims to reconstruct points clouds given rearranged point clouds as input. To this end, Sauder & Sievers (2019) equally split the 3D space into $k^3$ voxels, rearrange $k^3$ voxels and train a model to predict the original voxel label for each point. However, these random permutations destroy all spatial information that the model could have used to predict the true point cloud. Inspired by cover-trees (Beygelzimer et al., 2006), Sharma & Kaul (2020) utilised ball covers for hierarchical partitioning of points. They then train a model to classify each point to their assigned clusters. However, the selection of the ball centroids is somewhat random and they need to pre-train from scratch for each fine-tuning task. Instead, our method creates spatially realistic occlusions that a completion model learns to reconstruct. As such, this model learns how to naturally encode 3D object shape and contextual information. Recently there is a new method called PointContrast (Xie et al., 2020b) which mainly uses contrastive learning for pre-training indoor segmentation models. Our method is more general and transferable compared with theirs.

Point cloud completion (Yuan et al., 2018) has received attentions in recent years. Most works aim at achieving a lower reconstruction loss by incorporating 1) a better encoder (Xie et al., 2020a; Huang et al., 2020), 2) a better decoder (Tchapmi et al., 2019; Wen et al., 2020b); 3) cascaded refinement (Wang et al., 2020a) and 4) multi-viewed consistency (Hu et al., 2020b).

Completing 3D shapes for model initialisation has been considered before. Schönberger et al. (2018) used scene completion (Song et al., 2017; Dai et al., 2020; Hou et al., 2020) as an auxiliary task to initialise 3D voxel descriptors for visual localisation. They generated nearly-complete and partial voxelised scenes based on depth images and trained a variational autoencoder for completion. They have showed that the pre-trained encoder is more robust under different viewpoints and weather conditions. We adapt this idea to pre-training for point clouds. We have shown that our initialisation is better than random and prior methods in terms of 1) object understanding; 2) invariance under transformations; and 3) downstream task performance.

## 5 Discussion

In this work, we have demonstrated that why and how the Occlusion Completion (OcCo) learns the representations on point clouds that are more transformation invariant, more accurate in few-shot learning, and in various classification and segmentation fine tuning tasks, compared to prior work. In future, it would be interesting to design a completion model that is explicitly aware the view-point of the occlusion. A model like this would likely converge even quicker, and require fewer parameters, as this knowledge could act as a stronger inductive bias during learning. In general, we advocate for structuring deep models using graphical constraints as an inductive bias to improve learning.

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

# A  DESIGN OF THE COMPLETION MODEL

Previous point completion models (Dai et al., 2017b; Yuan et al., 2018; Tchapmi et al., 2019; Wang et al., 2020a) all use an "encoder-decoder" architecture. The encoder maps a partial point cloud to a vector of a fixed dimension, and the decoder reconstructs the full point cloud.

In the OcCo experiments, we exclude the last few MLPs of PointNet and DGCNN, and use the remaining architecture as the encoder to map a partial point cloud into a 1024-dimensional vector. We adapt the folding-based decoder design from PCN, which is a two-stage point cloud generator that produces a coarse and a fine-grained output point cloud $(Y_{coarse}, Y_{fine})$ for each input. We removed all the batch normalisation layers in the folding-based decoder since we find it brings negative effects in the completion process in terms of Chamfer distance loss and convergent speed. On the basis of prior self-supervised learning methods, SimCLR (Chen et al., 2020), MoCo (He et al., 2020) and BYOL (Guo et al., 2020), we find the batch normalisation is important in the encoder yet harmful for our decoder. Also, we find the L2 normalisation in the Adam optimiser is undesirable for completion training but brings improvements on the downstream fine-tuning tasks.

The predicted coarse point cloud $\hat{Y}_{coarse}$, which represents the global geometry of a shape, is generated via a set of fully connected layers. A folding-based generator is used to predict the local fine structures of each point in $\hat{Y}_{coarse}$, this results in $\hat{Y}_{fine}$. The folding based structures is proved to be good at approximating a smooth surface which reflects the local geometry. During training, $Y_{coarse}$ and $Y_{fine}$ are generated via randomly sampling 1024 and 16384 points from the mesh, respectively.

We use either Chamfer Distance (CD) or Earth Mover Distance (EMD) as the loss function for the completion model. We use a normalised and symmetric (thus commutative) version of Chamfer Distance (CD) to quantify the differences between two point clouds $\hat{P}$ and $P$:

$$\text{CD}(\hat{P}, P) = \frac{1}{|\hat{P}|} \sum_{\hat{x} \in \hat{P}} \min_{x \in P} ||\hat{x} - x||_2 + \frac{1}{|P|} \sum_{x \in P} \min_{\hat{x} \in \hat{P}} ||x - \hat{x}||_2. \tag{6}$$

Note that it is no need that the two point cloud $\hat{P}$ and $P$ have the same size. But when calculating the Earth Mover Distance (EMD), $\hat{P}$ and $P$ are usually required to have the same number of points:

$$\text{EMD}(\hat{P}, P) = \min_{\phi:\hat{P} \to P} \frac{1}{|\hat{P}|} \sum_{\hat{x} \in \hat{P}} ||\hat{x} - \phi(\hat{x})||_2, \tag{7}$$

where $\phi$ is a bijection between points in $\hat{P}$ and $P$. Note that EMD is not commutative. Since finding the optimal mapping $\phi$ is quite time consuming, we use its approximation form Bertsekas (1985).

The loss $l$ of the completion task is a adaptive weighted sum of coarse and fine generations:

$$l = d_1(\hat{Y}_{coarse}, Y_{coarse}) + \alpha * d_2(\hat{Y}_{fine}, Y_{fine}), \tag{8}$$

where the step-wise trade-off coefficient $\alpha$ incrementally grows during training. In our experiments, we find that even with approximation, it is still suboptimal to use EMD for $d_2$, since it is inefficient to solve the approximate bijection mapping $\phi$ for over 16k point pairs. We evaluate both 'EMD+CD' and 'CD+CD' combinations for the loss $l$. We have found that OcCo with 'EMD+CD' loss has achieved comparable performance yet longer time in the downstream classification tasks compared with the 'CD+CD'. We use 'CD+CD' as the loss function in the OcCo pre-training process described in Section. 3.1 in terms of simplicity and efficiency.

# B  QUALITATIVE RESULTS FROM OCCO PRE-TRAINING

In this section, we show some qualitative results of OcCo pre-training by visualising the input, coarse output, fine output and ground truth at different training epochs and encoders. In Figure. 6, Figure. 7 and Figure. 8, we notice that the trained completion models are able to complete even difficult occluded shapes such as plants and planes. In Figure. 9 we plot some failure examples of completed shapes, possibly due to their complicated fine structures, while it is worth mentioning that the completed model can still completed these objects under the same category.

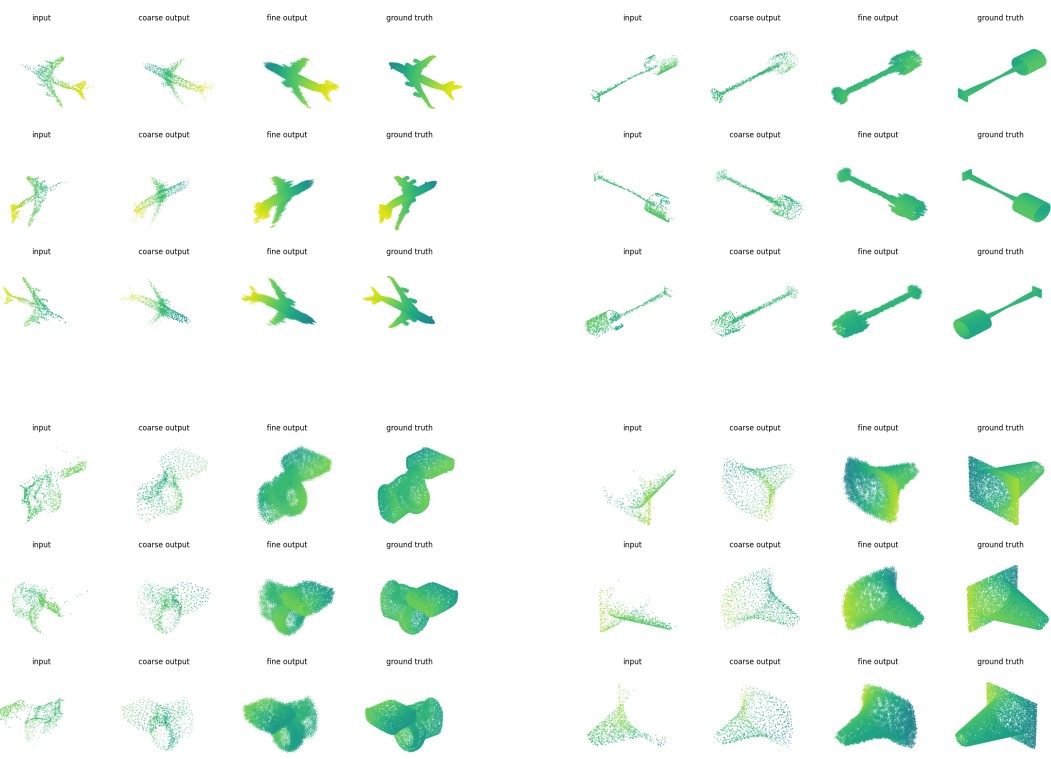

Figure 6: OcCo pre-training with PCN encoder on occluded ModelNet40.

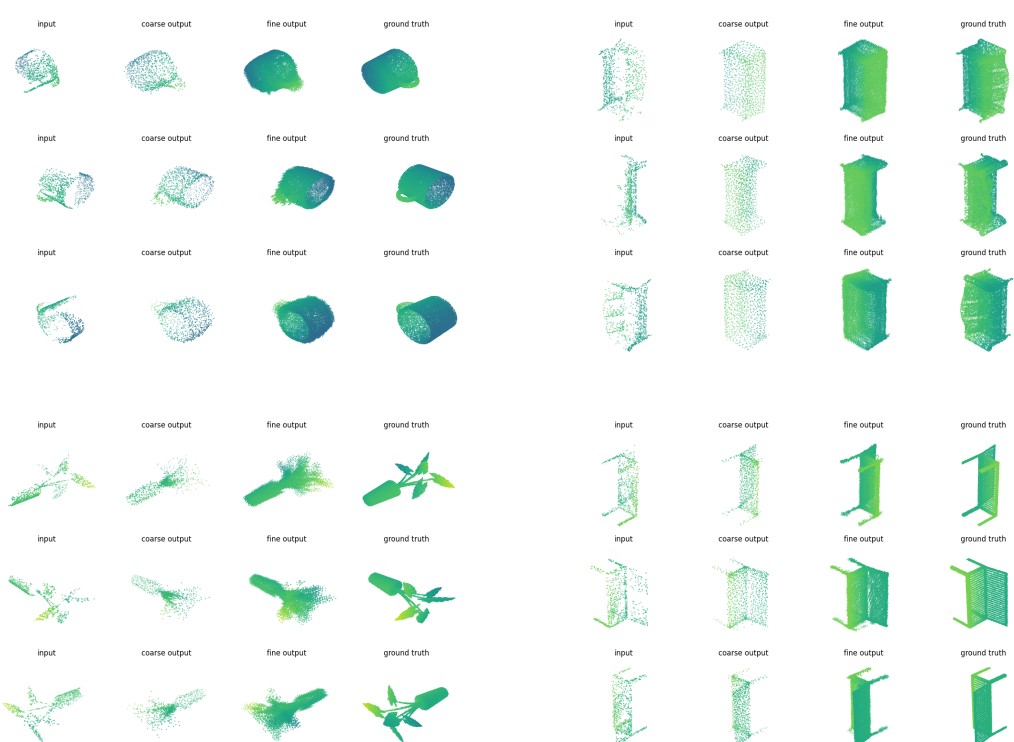

Figure 7: OcCo pre-training with PointNet encoder on occluded ModelNet40.

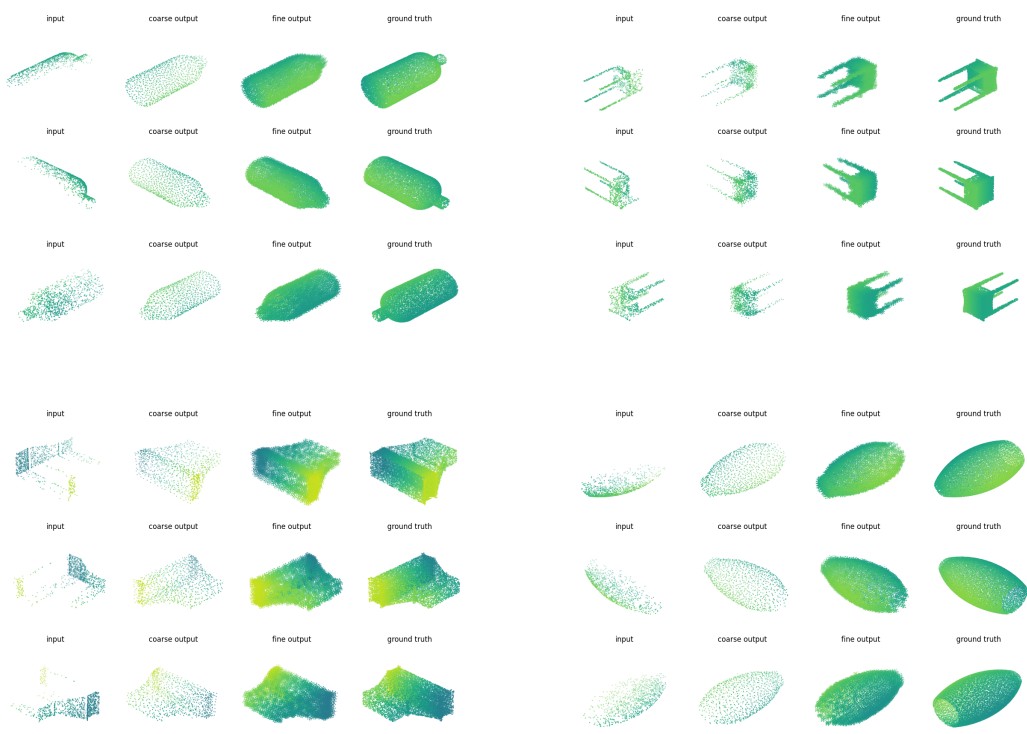

Figure 8: OcCo pre-training with DGCNN encoder on occluded ModelNet40.

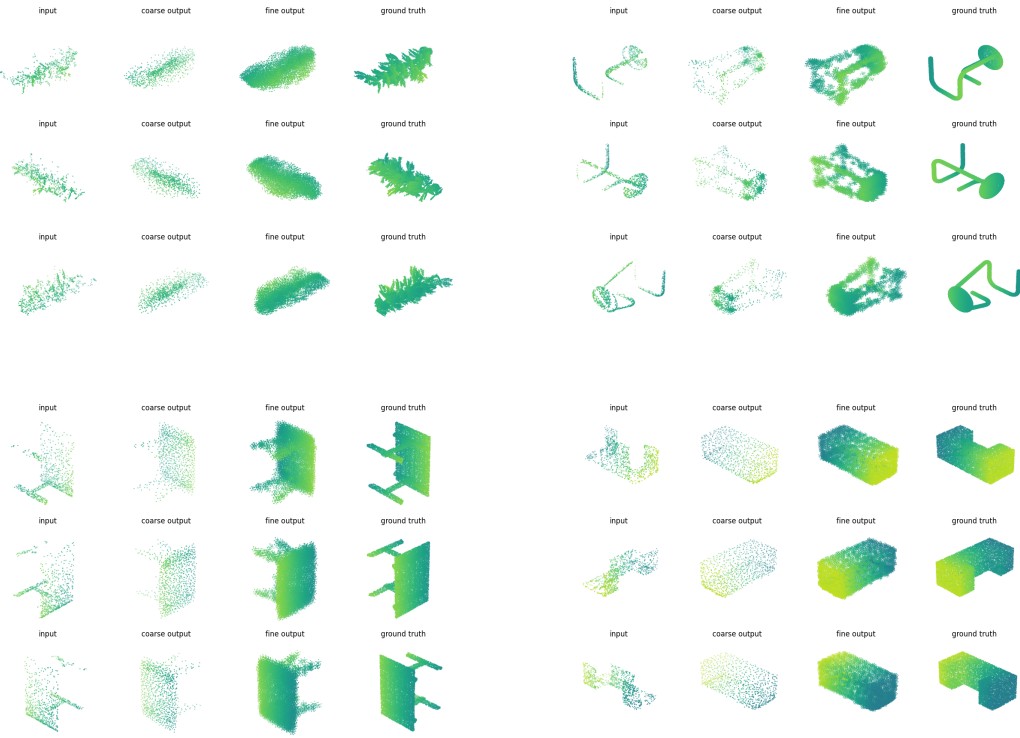

Figure 9: Failure completed examples during OcCo pre-training.

## C    VISUALISATION OF PART ACTIVATION MAP AND GROUND TRUTH

Figure 10: Learned features in OcCo pre-training

## D    ANALYSIS ON THE EFFECTS OF OCCO PRE-TRAINING DATASETS

We compare the occluded datasets based on ModelNet40 and ShapeNet8 for the OcCo pre-training. We construct the ModelNet Occluded using the methods described in Section 2 and for ShapeNet Occluded we directly use the data provided in the PCN, whose generation method are similar but not exactly the same with ours. Basic statistics of these two datasets are reported in Table 6. Compared with the ShapeNet Occluded dataset which is publicized by PCN and used in all the follow-ups(Tchapmi et al., 2019; Wang et al., 2020a), our occluded ModelNet dataset has more object categories, more view-points, more points per object and therefore is more challenging. We believe such differences will help the encoder models learn a more comprehensive and robust representation which is transferable to downstream tasks. To support our idea, we perform OcCo pre-training on these two datasets respectively, and test their performance on ModelNet40 and ShapeNet Occluded classification benchmarks.

Table 6: Statistics of occluded datasets for OcCo pre-training

| Name | # of Class | # of Object | # of Views | # of Points/Object |
|---|---|---|---|---|
| ShapeNet Occluded (PCN) | 8 | 30974 | 8 | 1045 |
| ModelNet Occluded (OcCo) | 40 | 12304 | 10 | 20085 |

The reason of choosing these two datasets for benchmarking is, ShapeNet Occluded is the out-of-domain data for the models pre-trained on ModelNet Occluded, and vice versa. We believe it will give us sufficient information on which occluded dataset should be preferred the OcCo pre-training. The Results are shown in Table 7.

Table 7: Performance of OcCo pre-trained models with different pre-trained datasets

| OcCo Settings | | Classification Accuracy | |
| --- | --- | --- | --- |
| Encoder | Pre-Trained Dataset | ModelNet Oc | ShapeNet Oc |
| PointNet | ShapeNet Oc | 81.0 | 94.1 |
| | ModelNet Oc | **85.6** | **95.0** |
| PCN | ShapeNet Oc | 81.6 | 94.4 |
| | ModelNet Oc | **85.1** | **95.1** |
| DGCNN | ShapeNet Oc | 86.7 | 94.5 |
| | ModelNet Oc | **89.1** | **95.1** |

From Table 7, we see that the OcCo models pre-trained on ShapeNet Occluded do not perform as well as the ones pre-trained on ModelNet Occluded in most cases. Thus in our experiments, we reports the results pre-trained on ModelNet Occluded.

By visualising the objects from the ShapeNet Occluded (in Figure. 11), we believe this performance deficiency in downstream fine-training of pre-trained models is due to the quality of the generated occluded point clouds (in comparison with our generated dataset shown in Figure. 2). Further, we think our dataset is a more challenging task for all the present completion models.

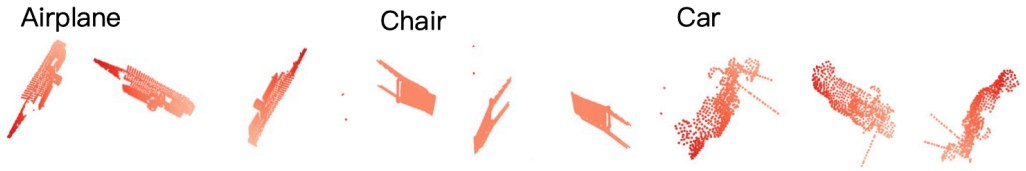

Figure 11: Examples from ShapeNet Occluded which fail to depict the underlying object shapes

# E    DETAILED RESULTS ON TRAINING A LINEAR SVM FOR CLASSIFICATION

We first described the benchmark datasets that are used for classification in Table 8.

Table 8: Statistics of classification datasets

| Name | Type | # Class | # Training | # Testing |
| --- | --- | --- | --- | --- |
| (MN40) ModelNet40 (Wu et al., 2015) | synthesized | 40 | 9,843 | 2,468 |
| (SN10) ScanNet10 (Dai et al., 2017a) | real scanned | 10 | 6,110 | 1,769 |
| (SO15) ScanObjectNN  (Uy et al., 2019) | real scanned | 15 | 2,304 | 576 |

To make a comprehensive and convincing comparison, we follow the similar procedures from (Achlioptas et al., 2018; Han et al., 2019; Sauder & Sievers, 2019; Wu et al., 2016; Yang et al., 2018), to train a linear Support Vector Machine (SVM) to examine the generalisation of OcCo encoders that are pre-trained on occluded objects from ModelNet40. For all six classification datasets, we fit a linear SVM on the output 1024-dimensional embeddings of the train split and evaluate it on the test split. Since Sauder & Sievers (2019) have already proven their methods are better than the prior, here we only systematically compare with theirs. We report the results[2] in Table 9, we can see that all OcCo models achieve superior results compared to the randomly-initialized counterparts, demonstrating that OcCo pre-training helps the generalisation both in-domain and cross-domain.

---

[2]In our implementation, we also provide an alternative to use grid search to find the optimal set of parameters for SVM with a Radial Basis Function (RBF) kernel. In this setting, all the OcCo pre-trained models have outperformed the random initialised and Jigsaw pre-trained ones by a large margin as well.

Table 9: linear SVM on the output embeddings from random, Jigsaw and OcCo initialised encoders

| Dataset | PointNet | | | PCN | | | DGCNN | | |
|---|---|---|---|---|---|---|---|---|---|
| | Rand | Jigsaw | OcCo | Rand | Jigsaw | OcCo | Rand | Jigsaw | OcCo |
| ShapeNet10 | 91.3 | 91.1 | 93.9 | 88.5 | 91.8 | **94.6** | 90.6 | 91.5 | 94.5 |
| ModelNet40 | 70.6 | 87.5 | 88.7 | 60.9 | 73.1 | 88.0 | 66.0 | 84.9 | **89.2** |
| ShapeNet Oc | 79.1 | 86.1 | 91.1 | 72.0 | 87.9 | 90.5 | 78.3 | 87.8 | **91.6** |
| ModelNet Oc | 65.2 | 70.3 | 80.2 | 55.3 | 65.6 | **83.3** | 60.3 | 72.8 | 82.2 |
| ScanNet10 | 64.8 | 64.1 | 67.7 | 62.3 | 66.3 | **75.5** | 61.2 | 69.4 | 71.2 |
| ScanObjectNN | 45.9 | 55.2 | 69.5 | 39.9 | 49.7 | 72.3 | 43.2 | 59.5 | **78.3** |

## F  RE-IMPLEMENTATION DETAILS OF "JIGSAW" PRE-TRAINING METHODS

In this section, we describe how we reproduce the 'Jigsaw' pre-training methods from (Sauder & Sievers, 2019). Following their description, we first separate the objects/chopped indoor scenes into $3^3 = 27$ small cubes and assign each point a label indicting which small cube it belongs to. We then shuffle all the small cubes, and train a model to make a prediction for each point. We reformulate this task as a 27-class semantic segmentation, for the details on the data generation and model training, please refer to our released code.

## G  MORE COMPARISONS

In Table 10, we compare OcCo with prior point-cloud-specific pre-training methods (Alliegro et al., 2020). Our method obtains the best results on all settings. These results confirm that the inductive bias learned by reconstructing occluded point clouds is stronger than one based in reconstructing permuted clouds (Alliegro et al., 2020; Sauder & Sievers, 2019). Specifically, we believe that because OcCo does not rearrange object parts but instead creates point clouds that resemble real-world 3D sensor occlusions, the initialization better encodes realistic object shape and context.

Table 10: Accuracy comparison between OcCo and prior pre-training baselines Alliegro et al. (2020) on 3D object recognition benchmarks. ModelNet40-20% means only 20% of training data are used.

| Baseline | Dataset | Rand | Alliegro et al. (2020) | OcCo |
|---|---|---|---|---|
| PointNet | ModelNet40 | 89.2 | 89.7 | **90.2** |
| | ModelNet40-20% | 82.9 | 83.1 | **83.6** |
| | ScanObjectNN (OBJ_BG) | 73.7 | 71.3 | **80.2** |

## H  LABELLED SAMPLE EFFICIENCY

We investigate whether OcCo pre-training can improve the labelled sample efficiency of downstream tasks. Specifically, we reduce the labelled samples to 1%, 5%, 10% and 20% of the original training set for the ModelNet40 object classification task, and evaluate on the full test set. As shown in Table 11, OcCo-initialized models achieve superior results compared to the randomly-initialized models, demonstrating that OcCo with in-domain pre-training improves labelled sample efficiency.

Table 11: Sample efficiency with randomly-initialized and OcCo-initialized models.

| Baseline | PointNet | | | PCN | | | DGCNN | | |
|---|---|---|---|---|---|---|---|---|---|
| | Rand | Jigsaw | OcCo | Rand | Jigsaw | OcCo | Rand | Jigsaw | OcCo |
| 1% | 56.9 | 55.7 | 58.1 | 57.8 | 59.6 | 60.4 | 60.0 | 59.9 | **60.5** |
| 5% | 73.9 | 74.3 | 74.9 | 73.2 | 75.8 | 76.7 | 79.4 | 79.2 | **79.7** |
| 10% | 80.6 | 81.3 | 81.1 | 81.1 | 82.1 | 82.6 | 84.4 | 84.4 | **84.5** |
| 20% | 83.6 | 84.2 | 84.2 | 83.6 | 84.2 | 84.4 | 86.5 | 86.7 | **87.2** |

# I  NETWORKS AND TRAINING SETTINGS OF PCN ENCODER

We sketch the network structures of PCN encoder and output layers for downstream tasks in Figure 12.

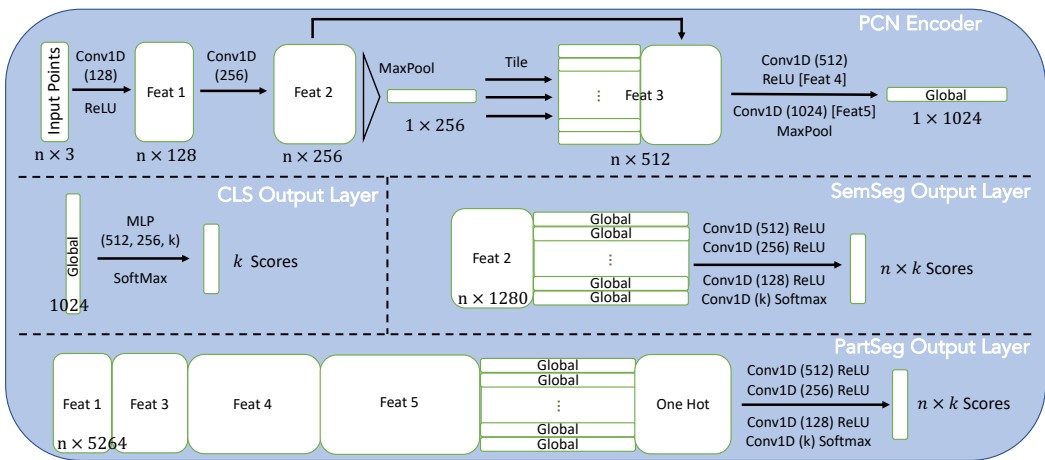

Figure 12: Encoder and Output Layers of PCN

# J  DETAILED RESULTS OF THE PART SEGMENTATION

Here in Table 12 we report the detailed scores on each individual shape category from ShapeNetPart, we bold the best scores for each class respectively. We show that for all three encoders, OcCo-initialisation has achieved better results over two thirds of these 15 object classes.

Table 12: Detailed Results on Part Segmentation Task on ShapeNetPart

| Shapes | PointNet | | | PCN | | | DGCNN | | |
|---|---|---|---|---|---|---|---|---|---|
| | Rand* | Jigsaw | OcCo | Rand | Jigsaw | OcCo | Rand* | Jigsaw* | OcCo |
| mean (point) | 83.7 | 83.8 | 84.4 | 82.8 | 82.8 | 83.7 | 85.1 | 85.3 | **85.5** |
| Aero | 83.4 | 83.0 | 82.9 | 81.5 | 82.1 | 82.4 | 84.2 | 84.1 | **84.4** |
| Bag | 78.7 | 79.5 | 77.2 | 72.3 | 74.2 | 79.4 | 83.7 | **84.0** | 77.5 |
| Cap | 82.5 | 82.4 | 81.7 | 85.5 | 67.8 | **86.3** | 84.4 | 85.8 | 83.4 |
| Car | 74.9 | 76.2 | 75.6 | 71.8 | 71.3 | 73.9 | 77.1 | 77.0 | **77.9** |
| Chair | 89.6 | 90.0 | 90.0 | 88.6 | 88.6 | 90.0 | 90.9 | 90.9 | **91.0** |
| Earphone | 73.0 | 69.7 | 74.8 | 69.2 | 69.1 | 68.8 | 78.5 | **80.0** | 75.2 |
| Guitar | 91.5 | 91.1 | 90.7 | 90.0 | 89.9 | 90.7 | 91.5 | 91.5 | **91.6** |
| Knife | 85.9 | 86.3 | 88.0 | 84.0 | 83.8 | 85.9 | 87.3 | 87.0 | **88.2** |
| Lamp | 80.8 | 80.7 | 81.3 | 78.5 | 78.8 | 80.4 | 82.9 | 83.2 | **83.5** |
| Laptop | 95.3 | 95.3 | 95.4 | 95.3 | 95.1 | 95.6 | 96.0 | 95.8 | **96.1** |
| Motor | 65.2 | 63.7 | 65.7 | 64.1 | 64.7 | 64.2 | 67.8 | **71.6** | 65.5 |
| Mug | 93.0 | 92.3 | 91.6 | 90.3 | 90.8 | 92.6 | 93.3 | 94.0 | **94.4** |
| Pistol | 81.2 | 80.8 | 81.0 | 81.0 | 81.5 | 81.5 | **82.6** | **82.6** | 79.6 |
| Rocket | 57.9 | 56.9 | 58.2 | 51.8 | 51.4 | 53.8 | 59.7 | **60.0** | 58.0 |
| Skateboard | 72.8 | 75.9 | 74.2 | 72.5 | 71.0 | 73.2 | 75.5 | **77.9** | 76.2 |
| Table | 80.6 | 80.8 | 81.8 | 81.4 | 81.2 | 81.2 | 82.0 | 81.8 | **82.8** |

# K    ALGORITHMIC DESCRIPTION OF OCCO

**Algorithm 1** Occlusion Completion (OcCo)

```
# P: an initial point cloud
# K: camera intrinsic matrix
# V: number of total view points
# loss: a loss function between point clouds
# c: encoder-decoder completion model
# p: downstream prediction model

while i < V:
   # sample a random view-point
   R_t = [random.rotation(), random.translation()]

   # map point cloud to camera reference frame
   P_cam = dot(K, dot(R_t, P))

   # create occluded point cloud
   P_cam_oc = occlude(P_cam, alg='z-buffering')

   # point cloud back to world frame
   K_inv = [inv(K), zeros(3,1); zeros(1,3), 1]
   R_t_inv = transpose([R_t; zeros(3,1), 1])
   P_oc = dot(R_t_inv, dot(K_inv, P_cam_oc))

   # complete point cloud
   P_c = c.decoder(c.encoder(P_oc))

   # compute loss, update via gradient descent
   l = loss(P_c, P)
   l.backward()
   update(c.params)
   i += 1

# downstream tasks, use pre-trained encoders
p.initialize(c.encoder.params)
p.train()
```

