# OpenReview forum: "Pre-Training by Completing Point Clouds"
_ICLR.cc/2021/Conference — Reject_

### Official Review · AnonReviewer1 · 2020-10-26
**A good one**

**Rating:** 7
**Confidence:** 4

**Review:**

This paper proposes a better pre-trained prior for a variety of downstream applications in point cloud analysis. The workflow of the pre-training mechanism is to first 1) generate occluded points that result from view occlusion and then 2) optimize the encoder to learn how to complete the occluded points from the partial point cloud. In downstream applications, the obtained encoder will be used as the initial weights in the network training. Empirical experiments have shown that such a pre-train mechanism can improve initialization over prior baselines and benefit a variety of tasks even with a large domain gap.

Pros:
1. The experimental results have shown a steady improvement in performance by using the proposed pre-training approach in different encoder architectures and different downstream applications. That provides strong support for validating the effectiveness of the proposed approach.
2. I also like the result that the initialization is only pre-trained on the occlusions generated from the ModelNet40 but still work in another dataset. And yet, the pre-training is done in a self-supervised manner. This is a great plus for this approach as it indicates that it could be a general-purpose booster for a wide range of applications without spending too much effort in collecting special-purpose dataset for pre-training.
3. The paper is well written and presented.

Cons:
1. The improvement, as shown in the statistics, is very incremental in most cases. I understand it is difficult to achieve better results on well-established benchmarks, but it somehow indicates the improvement is limited.
2. Though the paper already stated some nice explanation of the idea behind this approach, I would appreciate it if a more in-depth analysis of why such a pre-training mechanism could work is provided. Specifically, I would like more analysis of why such a pre-training method can adapt to different datasets? What are the common features that OcCo captures across different datasets?
Some visualization similar to Figure 3 would be helpful.

---- Final Rating ----

The authors' response has resolved my concerns. I would keep my positive rating.

---

> ### Author Response · Authors · 2020-11-22
> **Response to R1**
>
> Thank you for your encouraging words! To your questions:
>
> [..improvement seems limited..]
>
> Based on your comment we have run the most difficult classification and segmentation tasks over three runs and included standard deviations (Tables 3, 4, 5). We observe in nearly all cases OcCo outperforms other methods. Further, we also have new results for the setting of few-shot learning. Specifically, we compare 8 different models and 3 initialization methods. This includes a new point cloud pre-training method published at NeurIPS 2020 specifically designed for few-shot learning called cTree (Sharma & Kaul, 2020). The results (Table 2) show that models pre-trained with OcCo either outperform or have standard deviations that overlap with the best method in 7 out of these 8 settings, averaged over 10 runs.
>
> [..more analysis of why pre-training can adapt to different datasets..]
>
> Thank you for this suggestion we have included three additional analyses: 1. A clustering experiment to identify the number of concept detectors discovered by different pre-training methods (Figure 4); 2. Results on transformation invariance of the pre-trained representations (Table 1); 3. Additional visualization of feature activations (Figure 10, appendix). For 1 and 2 OcCo matches or outperforms current work. For 3 we see clear contiguous object parts, indicating the model has learned to identify key shapes. We believe this is why OcCo is so successful at adapting to new datasets: it is able to better respect the natural geometries of point clouds.

---

### Official Review · AnonReviewer3 · 2020-10-26
**paper shows promising results using point cloud completion tasks for pre-training representations, method is clearly described; however, highly related work not discussed that limits novelty of the proposed OcCo task, experimental setup not fully clear, making it hard to understand the results**

**Rating:** 7
**Confidence:** 4

**Review:**

The paper considers the problem of training networks for point cloud processing through a point cloud completion task. Given a point cloud, it is rendered from a set of viewpoints and for each viewpoint the set of visible points is determine. A network is then trained to generate the full point cloud from the partially observed point cloud for a given view. Here, an encoder-decoder architecture is used, where the encoder corresponds to the network that should be pre-trained. Experimental results show that the proposed method outperforms two baselines for three tasks (object classification, object part segmentation, and semantic segmentation), when using less training data, and that the pre-training on the occlusion task leads to faster convergence.

On the positive side, the occlusion completion (OcCo) task is clearly described and it should be fairly straightforward for a researcher to setup this task. The task requires no human annotation and is thus suitable for pre-training from large datasets captured in uncontrolled settings. The experiments cover a wide range of tasks and settings and show that the OcCo pre-training strategy outperforms random initialization and the approach from Saunders & Sievers. Here, the simplicity of the OcCo task coupled with its performance is clearly a major strength of the paper. In particular, Fig. 4 shows that the networks pre-trained on the OcCo task tend to converge much faster compared to the baselines.

On the negative side, I feel that the paper oversells the novelty of the OcCo task. The OcCo task is a variation of the (semantic) scene completion task that asks to complete a partial observation of a scene / object and is receiving attention in the computer vision community. Recent examples include [Dai et al.,  SG-NN: Sparse Generative Neural Networks for Self-Supervised Scene Completion of RGB-D Scans, CVPR 2020] and [Hou et al., RevealNet: Seeing Behind Objects in RGB-D Scans, CVPR 2020], with older works including [Firman et al., Structured prediction of unobserved voxels from a single depth image, CVPR 2016]. [Schönberger et al., Semantic Visual Localization, CVPR 2018] use (semantic) scene completion as a proxy task to train a 3D descriptors for 3D-3D matching between models. Given a voxelized partial observation of a scene, they train an encoder-decoder architecture to predict the complete volume (potentially also predicting semantic labels for each voxel). The embedding in the latent space are then used as 3D descriptors (i.e., the decoder part of the network is not needed at test time). In other words, they use the OcCo task for training their networks. They show that the learned representation generalizes between datasets and sensor modalities (training on 3D data obtained from stereo images, tested on LiDAR data). Given this result, I see limited novelty in using the OcCo task for pre-training point cloud networks and it does not seem very surprising that pre-training on the OcCo task should result in meaningful representations.

My second main point of criticism is the level of detail of the experimental evaluation. While the experiments cover a wide range of tasks and settings, I feel that crucial information needed to understand the results are missing:
1) Is the same dataset (ModelNet40) used to pre-train on the OcCo task also used to pre-train the JigSaw approach from Sauder & Sievers? Unfortunately, no details on the latter are provided in the main paper (or I was not able to find them), making it hard to understand how meaningful the comparison is.
2) There are no details on how the networks are trained for the different tasks, e.g., for how many epochs are the network trained for the task at hand?, do the networks converge for all pre-training strategies?
3) How significant are the improvements over the two baselines. For most considered settings, the improvements seem rather small, e.g., often less than 1 point compared to the Rand baseline in Tab. 2 and 3. Is this a meaningful improvment? Or would simply using a different random seed for training explain such a difference? Given that the paper claims that "These results demonstrate that the OcCo-initialized models have strong transfer capabilities on out-of-domain datasets" and that "OcCo-initialized models achieve superior results compared to the randomly-initialized models", this is an important question to answer.
I think there is enough space in the paper to include this information. Specifically, I do not think that Alg. 1 is necessary in the main paper (but would be good to have as an appendix) as the text and Fig. 1 already describe the approach in sufficient detail. Similarly, the z-Buffer algorithm is a classic computer graphics technique that is covered in basic lectures and does not need to be discussed in detail (e.g., see [Pittaluga et al., Revealing Scenes by Inverting Structure from Motion Reconstructions, CVPR 2019] briefly mentioning z-Buffering and the use of Delaunay triangulation for determining visibility). I think this space could be spend on providing more details.

In the current form, I do not think the paper is ready for publication as the paper, in my opinion, overclaims its contributions, misses relevant work (see also below), and misses crucial details necessary to understand the experimental results. As such, I am currently recommending to reject the paper. I believe that these issues can be addressed, but I would base my final recommendation based on the authors' feedback.

Here are additional detailed comments:
* In Sec. 2.1, I do not understand the comment "Our goal is to learn a randomized occlusion mapping o : P → P (where P is the space of all point clouds) from a full point cloud P to an occluded point cloud P". As far as I can tell, the mapping is not learned but follows a fixed pipeline.
* I don't understand how Eq. 2 "most closely approximates the inverse of eq. (1)". Eq. 2 is the inverse of Eq. 1. The only approximation that I could see if the 2D projection coordinates are rounded to the nearest integer.
* The introduction teases with the statement "Current 3D sensing modalities (i.e., 3D scanners, stereo cameras, lidars) have enabled the creation of large repositories of point cloud data (Rusu & Cousins, 2011; Hackel et al., 2017)." However, only synthetic data is used for pre-training, which is a bit disappointing. I am not convinced that the synthetic datasets "are qualitatively similar to point clouds in datasets where points are collected via 3D imaging devices such as handheld scanners (Dai et al., 2017a; Armeni et al., 2016) and lidar (Geiger et al., 2012)." Based on my experience, handheld scanners (RGB-D cameras or (multi-view) stereo cameras) produce much more noisy measurements with outliers while lidar sensors, especially for autonomous vehicles, typically produce sparser point clouds.
* Sec. 3.2 states that "We observe that, in early stage the encoder is able to learn low-level geometric primitives, i.e., planes, cylinders and cones, while later the network recognises more complex shapes like wings, leafs and upper bodies (non-rigid)." I am not sure how I see this in Fig. 3 since I don't know what the color-coding signifies.
* Looking at Fig. 3, I am not sure whether the statement that "clearly separable clusters are formed for different object classes" is true. There seems to be quite some overlap between classes, but this is also a bit hard to tell given the small size of the figure.
* [Yang et al., PointFlow: 3D Point Cloud Generation with Continuous Normalizing Flows, ICCV 2019] and [Gadelha et al., Label-Efficient Learning on Point Clouds using Approximate Convex Decompositions, ECCV 2020]  both propose generative models for point clouds. Both of them show that they can be used for unsupervised representation learning and they show competitive results for the task of only training an SVM classifier on top of the learned representation for ModelNet. Both should be discussed in the related work.

### After rebuttal phase ###
The comments by the authors and the revised version of the paper successfully address my concerns. I thus recommend to accept the paper.

---

> ### Author Response · Authors · 2020-11-22
> **Response to R3**
>
> [..related work to OcCo..]
>
> Thank you for these references, we have cited these in the related work section in our updated paper. While these methods and OcCo share important similarities, OcCo is different in a significant way: our approach only learns pre-trained weights on a single dataset (ModelNet40). We have shown that such pre-trained weights outperform all point cloud baselines in few-shot learning, improve generalization on many different classification tasks, and have better transformation invariance than prior work. As far as we are aware it was not known that such pre-training would yield such improvements across tasks and datasets.
>
> [..same dataset used to pre-train OcCo and Jigsaw?..]
>
> Yes, we use the same data (ModelNet40) for the pre-training for both OcCo adn Jigsaw methods.
>
> [..no details on how networks are trained..]
>
> In the original manuscript we included pre-training details in Section 3.1 and learning curves in Section 3.7 and Figure 4. We have added additional details in Section 3.1 in our updated manuscript.
>
> [..are improvements classification and segmentation significant?..]
>
> We have included new runs for both classification and segmentation across three runs in Sections 3.5-3.7. We report the mean and standard deviation of all methods and bold the best methods, and those that overlap the mean of the best in standard deviation. This shows that OcCo makes significant improvements over prior approaches.
>
> [..Alg 1, z-buffering unnecessary..]
>
> Thanks, we have moved these parts to the appendix. In place of these we have included new results on the abilities of our pre-trained representations including: detection of semantic concepts (Figure 4), transformation invariance (Table 1), as well as feature map visualization (Figure 10) in the appendix. We also have new results for the setting of few-shot learning. Specifically, we compare 8 different models and 3 initialization methods. This includes a new point cloud pre-training method published at NeurIPS 2020 specifically designed for few-shot learning called cTree (Sharma & Kaul, 2020). The results show that models pre-trained with OcCo either outperform or have standard deviations that overlap with the best method in 7 out of these 8 settings, averaged over 10 runs.
>
> [..isn’t o: P -> P not learned?..]
>
> Thank you for catching this, you are right this is not learned, we have corrected this.
>
> [..Eq. 2 is the inverse of Eq. 1..]
>
> This is correct, sorry for the confusing wording. We have modified the relevant descriptions in our methods.
>
> [..synthetic datasets aren’t that similar to handheld scanners and lidar..]
>
> We agree there are important differences, we have toned down the relevant arguments.
>
> [..what does the color mean in Fig. 3..]
>
> The color is based on the scalar value of that unit of each feature map. Since PointNet encoded each point feature separately, its major components are conv1d layers:
> $$
> \mbox{out}(N_i, C_j) = b_j + \begin{bmatrix}
> x_1, & y_1, & z_1 \\\\
> x_2, & y_2, & z_2 \\\\
>  & \cdots & \\\\
> x_N, & y_N, & z_N
> \end{bmatrix} \begin{bmatrix}
> w_{j1} \\\\
> w_{j2} \\\\
> w_{j3}
> \end{bmatrix} = \begin{bmatrix}
> x_1 w_{j1} + y_1 w_{j2} + z_1 w_{j3} + b_j \\\\
> x_2 w_{j1} + y_2 w_{j2} + z_2 w_{j3} + b_j \\\\
> \cdots \\\\
> x_N w_{j1} + y_N w_{j2} + z_N w_{j3} + b_j
> \end{bmatrix}
> $$
> Therefore it is quite convenient to track the features of each point. In our paper, “53rd Channel of Feat1” is essentially the 53th column of the 2D matrix named feature1 in Fig. 3.
>
> [..in Fig. 3 are clusters separated or overlapping?..]
>
> We agree this is not well-quantified so we have added clustering quality results based on the adjusted mutual information in Section 3.2.
>
> [..discuss related work Yang et al., Gadelha et al.]
>
> Thanks for the reminder, we have included these works in the updated pdf.
>
> [..I believe that these issues can be addressed, but I would base my final recommendation based on the authors' feedback..]
>
> Thank you for your careful review. Given these new results and responses we ask that you reconsider your score. Thank you for your consideration and time.

---

> > ### Comment · AnonReviewer3 · 2020-11-23
> > **Response to authors**
> >
> > Thank you very much for responding to my concerns and for the revised version of the paper.
> >
> > I think the revision significantly improves upon the original submission and addresses most of my comments. In particular, the newly added experiments more clearly show the advantages of the proposed approach over existing pre-training strategies. In addition, adding mean and standard deviations more clearly shows when the proposed approach outperforms existing strategies.
> >
> > On the negative side, it seems to me that the paper contradicts itself:
> > * The abstract and introduction claim that the paper introduces "a novel pre-training mechanism". Sec. 4.2 then states that "Completing 3D shapes for model initialisation has been considered before". Given that this approach has been used before, I would suggest to remove such claims and instead discuss the differences. At the moment, this discussion is rather generic. The paper claims that "OcCo is different in a significant way: our approach only learn pre-trained weights on a single dataset". Then again, the same can be said of Schönberger et al., who train on one dataset (using depth maps provided by stereo) and show that the learned representation generalizes to another dataset (where depth maps are provided by Lidar) without re-training or fine-tuning. in my opinion, the paper takes an existing idea (using completion as an auxiliary task that enables self-supervised learning), applies it on a new setting (pre-training for point clouds), and shows that this approach leads to improvements compared to existing strategies. I am not aware of prior work in this direction and think that the results shown in this paper are interesting for the community. I would personally prefer it if the paper was clearer earlier on about prior work using this task and I don't think that this would take away from the value of the paper.
> > * Sec. 4.2 states that "Yet, there is recent evidence that the representations learned by autoencoders are not necessarily good when used as initialization for downstream task". Yet, doesn't Yang et al., 2019b use an autoencoder and show that the learned representation generalizes to new tasks?
> >
> > Out of the two concerns above, I consider the first one more important and would like to see it resolved in a revised version of the paper. In the current form, I am still slightly leaning towards recommending rejection.

---

> > > ### Author Response · Authors · 2020-11-24
> > > **Follow up to R3**
> > >
> > > Thanks for your response! We have updated the manuscript again based on your suggestions.
> > >
> > > [..novelty statement contradiction, OcCo applies completion on a new setting..]
> > >
> > > Ah thanks for pointing this out! We have adjusted the relevant arguments in the abstract and the introduction, to make it consistent with the discussion in the related work. We have also added a brief discussion on the current progress in point completion.
> > >
> > > [..AE generalization, Yang et al., 2019b..]
> > >
> > > It seems that AE works (Yang et al., 2019b) and (Alberti et al., 2017), only investigate downstream performance by training a linear SVM to classify objects on the learned embeddings. AE methods have been recently outperformed by the Jigsaw completion of approach (Sauder & Sievers, 2019). We have changed the above sentence to clarify this nuance. Further we show that OcCo outperforms (Sauder & Sievers, 2019) on this task in appendix E.
> > >
> > > We would like to sincerely thank you again for helping us shape our work :) We’ve learned quite a lot through this, especially on scene completion!

---

> > > > ### Comment · AnonReviewer3 · 2020-11-24
> > > > **Follow up**
> > > >
> > > > Thank you very much for the revised version. This alleviates my main concerns and I am willing to change my recommendation to an acceptance rating.

---

### Official Review · AnonReviewer4 · 2020-10-27
**Limited novelty and weak improvements**

**Rating:** 4
**Confidence:** 4

**Review:**

The authors propose completing an occluded point cloud as a pretraining step for point cloud processing methods. Multiple occlusions are generated for the network to complete by simulating a camera perspective.

Pros:
- First work analyzing this specific pretraining for point clouds

Cons:
- Weak novelty
- Limited experimental reliability

Overall, the novelty of the paper seems rather weak. The only novel contribution is the idea of point cloud completion as a pretraining task. This idea is rather simple and similar techniques are well known and used in other fields such as NLP. Hence, it does not represent a significant methodological advancement.
Nevertheless, the paper would still be interesting if it showed extraordinary results in this particular field of application. Unfortunately, the experimental results seem weak as well. They show modest gains with respect to existing techniques and they lack any information on run-to-run variance. This makes it impossible to understand if the gains that are shown are statistically significant or just lucky runs with careful parameter tuning.

---

> ### Author Response · Authors · 2020-11-22
> **Response to R4**
>
> [..no information on run-to-run variance..]
>
> We have included new runs for both classification and segmentation across three runs (Tables 3, 4, 5). We report the mean and standard deviation of all methods and bold the best methods, and those that overlap the best in standard deviation. This shows that OcCo significantly matches or outperforms all baseline approaches.
>
> We believe that the improvements would be even larger, except that if a model has enough data for a task, it can converge to a point where it has largely overwritten its initial weights. To test this hypothesis, we have included new results on 8 few-shot learning tasks. We compare 8 different models and 3 initialization methods. This includes a new point cloud pre-training method published at NeurIPS 2020 specifically designed for few-shot learning called cTree (Sharma & Kaul, 2020). The results show that models pre-trained with OcCo either outperform or have standard deviations that overlap with the best method in 7 out of these 8 settings, averaged over 10 runs.
>
> [..paper would still be interesting if it showed extraordinary results..]
>
> Given these new results (and other new analyses in Section 3.2), which we believe are extraordinary for the few-shot setting (in Section 3.6), we ask that you reconsider your score. Thank you for your consideration and time.

---

### Official Review · AnonReviewer2 · 2020-10-28
**Since the idea itself is simple enough, the reviewer would not argue too much about the technical contribution of this paper, but concerns more about the analytic contributions. The reviewer’s concerns lie as follows.**

**Rating:** 5
**Confidence:** 5

**Review:**

The idea of this paper is simple but fascinating. Actually there are many studies concerning the task of point cloud completion, but using it as the initialization approach to improve the other tasks is quiet novel. The experimental results seem solid and quantitatively prove the effectiveness of the OcCo-initialization.

Since the idea itself is simple enough, the reviewer would not argue too much about the technical contribution of this paper, but concerns more about the analytic contributions. The reviewer’s concerns are as follows.
1. In addition to verify the effectiveness of OcCo-initialization, more analysis on why this simple idea can take effect should also be given. For example, the author should go deeper to explain why OcCo-initialized PointNet can outperform the random initialized PointNet (e.g. by visualizing the learned features of the two kinds of PointNet, like Figure 3).
2. The idea of OcCo-initialization can be concluded as some kind of task oriented initialization approach. Considering the simplicity of this idea, similar initialization strategy can be formulated, such as pre-training network on segmentation task and apply them on classification task. So why the author only chooses the completion task as the initialization strategy, or if task oriented initialization can be considered as a universal strategy in point cloud processing?
3. Although the experimental results look solid, the reviewer still concerns if the proposed OcCo-initialization can achieve the SOTA results or close enough to the current SOTA. For example, PointCNN can achieve much better segmentation results compared to the methods in Table 3. So if the OcCo-initialization can still succeed in the PointCNN which has better ability of learning point cloud features?
4. The author is advised to clarify the necessity of Sec 2.1, which is the main part of the model description. In reviewer’s opinion, a method to generate partial point cloud from single view is essentially not a technical contribution for this paper.

---

> ### Author Response · Authors · 2020-11-22
> **Response to R2**
>
> [..more analysis, e.g., visualization..]
>
> Thank you for this suggestion. We have included more analysis on what has been learned in OcCo pre-training compared to prior work. This includes Feature Visualization (Olah et al., 2017), Network Dissection (Bau et al., 2017), Clustering metrics (NMI, AMI) on Clustering of Embeddings under SO(3) transformation. Please see section 2.3 and 3.2 in the updated file.
>
> [..why not pre-train on segmentation and apply to classification?..]
>
> Thank you for raising this point. One of the biggest benefits of our proposal to use completion as a pre-training method is it does not require any labeled data. While one could pre-train a network on segmentation and apply this to a classification task, it requires labels in both settings, which is limited. In our case, we are able to leverage large unlabeled point cloud datasets which are growing rapidly due to the increase LIDAR and similar sensors generating point clouds.
>
> [..can OcCo improve PointCNN?..]
>
> We have strong evidence that it will. We’ve been able to show that our OcCo initialization consistently improves over random initialization for three popular point cloud models: PointNet, PCN, DGCNN. So we have strong evidence that it would also improve the performance of PointCNN. Further, we have added new results on 8 different few-shot learning settings. This comparison includes PointCNN (and 7 other point cloud models). We also compare with a new pre-training method published at NeurIPS 2020 specifically designed for few-shot learning called cTree (Sharma & Kaul, 2020). The results show that models pre-trained with OcCo either outperform or have standard deviations that overlap with the best method in 7 out of these 8 settings, averaged over 10 runs.
>
> [..Sec 2.1 can be shortened..]
>
> Thanks for this. We agree, our goal in Section 2.1 was to make our method more transparent to readers unfamiliar with stereo vision. We have shortened and moved parts of this section to the appendix.

---

### Author Response · Authors · 2020-11-22
**Overall Response**

We thank all the reviewers for the time and comments!

We have brought a major update for our submission on:
i) Analysis on learned features in OcCo pre-training (in Sections 3.2, 3.3), using Feature Visualization (Olah et al., 2017), Network Dissection (Bau et al., 2017) and Clustering metrics (NMI, AMI) on the learned embeddings under SO(3) transformation. Our method shows superior performance over baselines under the transformation.
ii) Few-shot learning (in Section 3.4), where we compare OcCo with a new baseline (Sharma & Kaul, 2020), as well as prior baselines. OcCo beats or matches the best method on 7 of the 8 settings.
iii) Ran the hardest fine tuning classification and segmentation tasks (due to computational constraints) for three separate runs (Tables 3, 4, 5). OcCo shows meaningful improvements overall baselines.

We respond to each reviewers comments in detail below.

---

### Decision · Program_Chairs · 2021-01-07
**Final Decision**

**Decision:**

Reject

**Comment:**

The paper proposes an unsupervised pretraining approach for 3D recognition, which is based on point cloud completion. The initial review receives a mixed rating, with two reviewers rate the paper below the bar and two above the bar. After the rebuttal, R3 changes the opinion from above the bar to a rejection recommendation. While several reviewers recognize the simplicity of the proposed method, R2 and R4 consider the proposed method a straightforward extension of known approaches for NLP and vision tasks. A lack of novelty was also pointed out as a weakness by R3 and R4. After consolidating the reviews and the rebuttal, the AC finds the weakness claims convincing and determines the paper is not ready for publication in the current form.